# The Role of Myeloid-Derived Suppressor Cells (MDSCs) in the Development and/or Progression of Endometriosis-State of the Art

**DOI:** 10.3390/cells10030677

**Published:** 2021-03-18

**Authors:** Dorota Suszczyk, Wiktoria Skiba, Joanna Jakubowicz-Gil, Jan Kotarski, Iwona Wertel

**Affiliations:** 1Independent Laboratory of Cancer Diagnostics and Immunology, Department of Oncological Gynaecology and Gynaecology, Medical University of Lublin, Staszica 16, 20-081 Lublin, Poland; dorotasuszczyk@umlub.pl (D.S.); wiktoriaskiba@umlub.pl (W.S.); 2Department of Functional Anatomy and Cytobiology, Maria Curie-Sklodowska University, Akademicka 19, 20-033 Lublin, Poland; jjgil@poczta.umcs.lublin.pl; 3Department of Gynaecologic Oncology and Gynaecology, Medical University of Lublin, Staszica 16, 20-081 Lublin, Poland; jankotarski@umlub.pl

**Keywords:** myeloid-derived suppressor cells (MDSCs), endometriosis, endometriosis-associated ovarian cancer (EAOC), immunosuppression, inflammation, microenvironment

## Abstract

Endometriosis (EMS) is a common gynecological disease characterized by the presence of endometrial tissue outside the uterus. Approximately 10% of women around the world suffer from this disease. Recent studies suggest that endometriosis has potential to transform into endometriosis-associated ovarian cancer (EAOC). Endometriosis is connected with chronic inflammation and changes in the phenotype, activity, and function of immune cells. The underlying mechanisms include quantitative and functional disturbances of neutrophils, monocytes/macrophages (MO/MA), natural killer cells (NK), and T cells. A few reports have shown that immunosuppressive cells such as regulatory T cells (Tregs) and myeloid-derived suppressor cells (MDSCs) may promote the progression of endometriosis. MDSCs are a heterogeneous population of immature myeloid cells (dendritic cells, granulocytes, and MO/MA precursors), which play an important role in the development of immunological diseases such as chronic inflammation and cancer. The presence of MDSCs in pathological conditions correlates with immunosuppression, angiogenesis, or release of growth factors and cytokines, which promote progression of these diseases. In this paper, we review the impact of MDSCs on different populations of immune cells, focusing on their immunosuppressive role in the immune system, which may be related with the pathogenesis and/or progression of endometriosis and its transformation into ovarian cancer.

## 1. Introduction

Endometriosis (EMS) is known as the most common gynecological disease. It affects approximately 10% of women of reproductive age and is connected with the presence of endometrial tissue outside the uterus. The disorder was recognized as a benign condition for many years, but now it is classified as a tumor-like lesion [1].

Various investigations have provided data supporting the observation that endometriosis may transform into endometriosis-associated ovarian cancer (EAOC). As demonstrated in several studies, the possible factors contributing to the dysfunctions include pelvic inflammatory diastases, age, life in highly urbanized cities, depression, and childlessness [2]. Genetic mutations in *ARID1A*, *K-RAS*, *PTEN*, and *β-catenin/Wnt* and microsatellite instability are also considered as important factors for the progression of endometriosis to EAOC [3]. It has been demonstrated that endometrial implants have similar characteristics to those of ovarian cancer (OC), i.e., potential to invade surrounding tissue, neoangiogenesis, reduced ability to undergo apoptosis, and local inflammation [4].

Identification of pathways associated with the transformation from endometriosis to cancer is an area of intensive research. However, the underlying mechanisms are still unclear [5].

Recent studies have also shown that endometriosis is associated with changes in the systemic and local immunity. The underlying mechanisms include quantitative and functional disorders of neutrophils, monocytes/macrophages, dendritic cells (DCs), natural killer cells (NK), and T cells [6]. A few reports have shown that immunosuppressive cells such as regulatory T cells (Tregs) and myeloid-derived suppressor cells (MDSCs) may promote the progression of endometriosis [7,8,9,10].

Taking into account the complex nature of the inflammatory milieu, in this paper, we review the impact of MDSCs on different populations of immune cells, focusing on their immunosuppressive role in the immune system. We also discuss MDSCs activity which may be associated with the pathogenesis and/or progression of endometriosis.

## 2. Immunophenotype of Myeloid-Derived Suppressor Cells (MDSCs)

Myeloid-derived suppressor cells represent a heterogeneous group of immune cells that can suppress anti-tumor immunity. This population includes immature myeloid cells at different stages of development such as dendritic cells, granulocytes, and macrophage precursors [11]. In fact, the population of MDSCs can be a mixture of immature and mature cells [12]. MDSCs are divided into three major subsets: monocytic (M-MDSCs), polymorphonuclear (PMN-MDSCs), and early stage MDSCs (eMDSCs) [13,14,15]. The appearance of this tolerogenic population is a common trait of cancer and other noncancerous diseases, such as sepsis and bacterial, viral, and parasitic infections, chronic inflammations, and autoimmune diseases [16,17,18]. Recent studies have demonstrated the role of MDSCs in aging, obesity, transplantation, and pregnancy [15].

The activity of MDSCs in pathological conditions is multidimensional: they can inhibit specific antitumor immune response, secrete immunosuppressive factors, and generate an inflammatory microenvironment [12]. The presence of MDSCs has an impact on the efficacy of immunotherapies and patient outcomes. The latest literature emphasizes the role of vascular endothelial growth factor A (VEGF-A). VEGF-A secreted by cancer cells not only induces angiogenesis, but also causes immunosuppression. VEGF blocks DC maturation causing decreased antigen presentation to T cells. This mechanism of angiogenesis-directed immune tolerance includes accumulation of immunoregulatory cells, e.g., Treg cells and MDSCs, and inhibition of T cell differentiation, proliferation, and functions. Since VEGF has such a great impact on angiogenesis and immunosuppression, therapy with VEGF blockade is another potential method to increase the anti-tumor activity of immunotherapy [19].

The migration of MDSCs into tumors and inflamed tissues is stimulated by several chemokines such as CXCL1, CXCL2, and CXCL5 by binding the receptor CXCR2 [18]. MDSCs expressing CXCR2 are considered as promoters of metastasis, T cell exhaustion, and tumor cell expansion in breast cancer [20]. They inhibit the efficacy of the immune system cells inducing immunosuppression and/or anergy of NK and T cells. Relationships between the response to programmed cell death protein 1 (PD-1) and cytotoxic T lymphocyte-associated protein 4 (CTLA-4) checkpoint inhibitors and the presence of MDSCs have been reported in clinical trials [21]. Blocking the CXCR2 receptor in pancreatic cancer leads to an elevated concentration of T cells within the tumor microenvironment (TME), reduced metastasis, and enhanced response to anti-PD-1 therapy. Interestingly, tumor-infiltrating MDSCs have higher immunosuppressive activity than MDSCs form peripheral blood (PB) [21]. Despite the same phenotype, MDSCs originating from healthy donors are not as immunosuppressive as MDSCs from pathological tissues. Moreover, it has been reported that their suppressive activity is limited by the inflammatory milieu. It was shown that monocytes from healthy subjects are able to acquire the phenotype and suppressive function of M-MDSCs under exposure to cancer cells and the specific microenvironment with a high level of interleukin 10 (IL-10) or prostaglandin E2 (PGE2). A similar mechanism is observed in the transformation of neutrophils into PMN-MDSCs [12].

Each MDSC population has a unique phenotype and morphological and functional characteristics [13,14,22,23].

### 2.1. Monocytic MDSCs (M-MDSCs)

M-MDSCs are detectable in a small amount in the peripheral blood of patients without disturbances. Their phenotype can be defined as CD11b^+^ (or CD33 instead of CD11b) HLA-DR^−^CD14^+^CD15^−^ cells [13]. In the cancer microenvironment, M-MDSCs can differentiate into inflammatory dendritic cells (inf-DCs) and tumor-associated macrophages (TAMs) [24]. The presence of M-MDSCs in PB is connected with a shorter progression-free interval (PFI) in different cancers, including bladder, colorectal, non-small-cell lung carcinoma, thyroid, and uterine cancer. Moreover, an elevated number of M-MDSCs is responsible for failure in chemotherapy in breast, cervical, colorectal, and prostate cancer. M-MDSCs are potent immunosuppressive cells (more effective than PMN-MDSCs) [24].

### 2.2. Polymorphonuclear MDSCs (PMN-MDSCs)

PMN-MDSCs are morphologically similar to neutrophils. These cells are not detectable in the peripheral blood of healthy subjects. They express CD11b^+^CD15^+^(CD66b^+^) markers, whereas CD14 is not expressed on their surface. In recent studies, lectin-type oxidized LDL receptor 1 (LOX-1) was used to distinguish PMN-MDSCs from neutrophils. The most important effect of PMN-MDSCs is inhibition of the activity of T cells, myeloid cells, and NK cells. It is known that they are involved in regulation of the cell cycle, G-protein signaling, the cAMP response element binding protein (CREB) pathway, and autophagy. In cancers, PMN-MDSCs are implicated in tumor growth, angiogenesis, and metastasis [18]. The most interesting feature is the ability of PMN-MDSCs to convert senescent cancer cells into proliferating cancer cells [13,18].

## 3. Immunosuppressive Activity of MDSCs in the Immune System

MDSCs interact with many populations of immune cells in the organism, e.g., monocytes/macrophages, NK cells, and lymphocytes T and B.

### 3.1. MDSCs Influence Monocytes/Macrophages

Myeloid-derived suppressor cells and tumor-associated macrophages (TAMs) are the most widely present tumor-infiltrating immune cells belonging to a larger population of cells, i.e., tumor-associated myeloid cells (TAMCs) [25]. M-MDSCs and monocytes/macrophages are attracted to the tumor niche using CCL2/CCR2 and colony stimulating factors (CSF) pathways. Multiple factors secreted by tumor cells contribute to the differentiation of MO/MA into TAMs [25]. Interestingly, MDSCs have also been shown to affect MO/MA and thus attenuate innate immunity and promote tumor progression [26]. This cell contact-dependent cross-talk between MDSCs and macrophages, supported by IL-1β induced inflammation [27], enhances the pro-tumor activity of the latter group and increases the production of IL-10 by MDSCs through a TLR4-mediated mechanism [26]. Macrophages initially are M1-like MA known as antitumor cells characterized by high expression of IL-12 and low expression of IL-10 [28]. Via IL-10 production, MDSCs cause downregulation of M1-like IL-12 production. Additionally, MDSC-MA interaction results in decreased MHC class II macrophage expression responsible for the antigen presenting capacity of these cells. Interestingly, M1 cells convert into pro-tumor M2 macrophages [29,30,31,32], which unlike M1 produce high levels of IL-10 and low levels of IL-12 [27].

Furthermore, MDSCs not only produce IL-10 but also stimulate M2-like macrophages that produce high amounts of IL-10 [33,34]. Thus, MDSCs promote the accumulation and activation of immunosuppressive M2 macrophages [34]. However, Cassetta et al. [35] have shown in their research that there are differences in the gene expression profile between tumor-resident macrophages and tumor-associated MA and that TAMs cannot always be assigned as the pure M1 or M2 phenotype. Despite the high degree of MA plasticity and heterogeneity, they are broadly grouped based on their activation status, namely M1 and M2 [36], and have different phenotypes, contrasting functions, and most importantly different secretion profiles [37]. M1 macrophages are known for their antitumor activity, which they owe to the expression of proinflammatory and immunostimulatory effector molecules and promotion of the Th1 immune response. In contrast, M2 cells possess tumor promoting activity, express a broad series of anti-inflammatory effector molecules, and promote Th2 response [36] as well as angiogenesis and immunotolerance. They also show profibrotic activity [38].

Lagana et al. [38] showed an increased number of both M1 and M2 macrophages in ovarian endometriomas in patients with stage I to IV of endometriosis. Interestingly, there was an upward trend in the percentage of the M2 population from stage I to IV and a reverse trend in the case of the M1 cell phenotype. These findings agree with other reports on the importance of the M1 to M2 switch both in mouse models and in endometriosis in humans [39]. Other reports based on a mouse model of endometriosis indicate the importance of endogenous macrophages in tissue remodeling in the process of endometriosis development and the necessity of the M1/M2 phenotype switch to growth in the ectopic lesion [40].

Moreover, MDSCs not only affect the activity of macrophages but can also become one of these cells. Corzo et al. showed M-MDSCs within hypoxic regions of solid tumors, mostly via the hypoxia inducible factor 1 (HIF) pathway, at a great rate differentiated into TAMs, i.e., a myeloid cell population that promotes tumor progression [26,36,41].

### 3.2. MDSCs Influence NK Cells

Natural killer cells are critical to the innate immune system and represent the first line of defense against transformed cells. MDSCs are responsible for reduction of the cytotoxic anti-tumor activity of NK cells by producing TGF-β in different types of tumors. TGF-β is a multifunctional factor, which reduces the proliferation of NK cells and inhibits the expression of natural cytotoxicity triggering receptor 3 (NKp30, CD247) and the natural killer group 2D receptor (NKG2D). In 2019, Huang et al. showed that TGF-β promoted escape of leukemia cells from host immunity. This process was connected with downregulation of target-NK signaling by suppressing CD48 expression. CD48 is known as a B-lymphocyte activation marker (BLAST-1) expressed on the surface of NK cells (and other immune cells) and taking part in adhesion and activation pathways [42,43].

Moreover, MDSCs produce NO, which restricts the cytotoxicity of NK cells connected with antibody-dependent cellular cytotoxicity (ADCC) by impoverishing the Fc receptor function. Reduced secretion of IFN-γ and tumor necrosis factor alpha (TNF-α) was observed in a co-culture of MDSCs and NK cells. Interestingly, increased production of indoleamine 2,3-dioxygenase (IDO) by MDSCs reduces proliferation and activation of NK cells and is connected with lower expression of natural cytotoxicity receptors (NCR), NKG2D, and DNAX accessory molecule-1 (DNAM-1) on their surface [42].

### 3.3. MDSCs Influence T Cell Activity

MDSCs have an impact on the activity of effector T cells via different mechanisms. One of them is involved in upregulation of arginase-1 (ARG-1) expression and leads to arginine depletion. ARG-1, i.e., an enzyme in the urea cycle, is responsible for converting L-arginine into urea and L-ornithine, which is essential to activate T cells. The translational blockade of T cells and cell cycle arrest in G0-G1 is a result of L-arginine depletion [44,45].

Another mechanism is based on the stimulation of inducible NO synthase (iNOS) and an impact on the production of nitric oxide (NO) and reactive oxygen species (ROS). The higher production of NO by MDSCs suppresses T cell function via inhibition of the phosphorylation of Janus kinase 3 or STAT5 or via induction of T cell apoptosis. MDSCs produce ROS and peroxynitrite (PNT), which hampers recognition of antigens by T cells [46]. MDSCs can nitrate T-cell receptor (TCR) proteins, which generates dissociation of the TCR complex and leads to the inhibition of TCR signaling. Additionally, MDSCs secrete transforming growth factor beta (TGF-β), which inhibits differentiation of Th1 and Th2 cells, reduces the cytotoxicity of CD8+ T cells, and suppresses the differentiation and responses of Th17 cells [46,47]. Furthermore, TGF-β stimulates the differentiation and migration of immunotolerant Treg cells [46]. MDSCs not only promote the expansion of natural Tregs but also induce Tregs through the secretion of interferon gamma (IFN-γ), TGF-β, and IL-10, and via CD40-CD40L interactions [26,48].

### 3.4. MDSCs Influence B Cell Function

B cells are an important compartment of humoral response responsible for recognition and presentation of antigens, regulation of the activity of T cells, and innate immunological response. They can be divided into four subpopulations: CD19^+^CD20^+^CD27^−^CD95^−^CD138^−^ naïve B-cells, CD19^+^CD20^+^CD27^+^CD95^+^CD138^−^ long-lived memory B-cells, CD19^+^CD20^−^CD27^+^CD95^+^CD138^+^ long-lived plasma cells, and regulatory Breg cells. It is difficult to describe the Breg phenotype due to the differences between Breg markers recognized in murine models and in different cancers [49,50].

MDSCs are able to regulate B-cell homeostasis (proliferation and antibody production) in a dose- and B cell stimulus-dependent manner. In 2017, Lelis et al. showed that PMN-MDSCs reduced the proliferation of B cells by cell-to-cell contact and induced death of B-cells. They also observed lower expression of co-stimulatory molecule CD86, whose level is high on activated B cells, and found that PMN-MDSCs were involved in the B-cell activation pathways. Additionally, PMN-MDSCs are able to inhibit IgM antibody production and proliferation by releasing ARG-1, NO, and ROS [51].

Kennedy and Knight (2015) have reported that MDSCs produce IL-1, which inhibits differentiation of progenitors into B cells [52]. It has also been shown that M-MDSCs secrete ARG-1, NO, ROS, peroxynitrite, PGE2, and TGF-β, which inhibit the production of IgG and IgM antibodies and proliferation of B cells. Furthermore, MDSCs exert an impact on the increased expansion of Breg cells. Bregs induce immune tolerance by production of IL-10, TGF-β, and IL-35. In this way, they exert an effect on the function of antigen presenting cells (APCs), T cells, and NK cells. It has been demonstrated that Bregs use a variety of mechanisms to induce immunotolerance; they suppress the activation of B cells by inhibiting CD4^+^T helper cells (Th cells), suppress Ig production, and increase the number of Tregs [46]. An elevated percentage of B cells was detected in the peritoneal cavity and peripheral blood of patients with endometriosis [53]. Nevertheless, the infiltration of CD19^+^ or CD138^+^ B cells was associated with poor prognosis in patients with ovarian cancer, including endometriosis-associated ovarian cancer [49,50]. However, the role of B cells in these disorders is not well understood.

It has been shown that abnormally activated B lymphocytes may be involved in induction of autoimmune response in patients suffering from endometriosis [54]. In particular, antigen presenting cells (APC), which present endometrial self-antigens to autoreactive T and B cells, play an important role in this response. Interestingly, an increased level of anti-endometrial antibodies has been detected in the serum of EMS patients [55,56]. Moreover, studies have shown that women with endometriosis have a greater risk of several autoimmune diseases than EMS-unaffected women [54,55]. Recent studies have demonstrated an association between endometriosis and increased risk of some autoimmune diseases, e.g., systemic lupus erythematosus (SLE), Sjogren’s syndrome (SS), celiac disease (CLD), multiple sclerosis (MS), and inflammatory bowel disease (IBD). The presence of endometriosis may be also related to co-occurrence of rheumatoid arthritis (RA), autoimmune thyroid diseases (ATD), and Addison’s disease [55]. However, it is unknown whether EMS is a risk factor or a consequence of these autoimmune disorders or whether they share the same mechanisms and biological pathways influencing their co-occurrence [55]. Therefore, further studies are recommended to explore this issue.

## 4. Relation between MDSCs and the Progression of Endometriosis—State of the Art

Chronic inflammation is one of the hallmarks of endometriosis. An increased level of cytokines (IL-1, IL-4, IL-6, IL-8, IL-10, IL-25, IL-37), growth factors like TNF-α and TGF-β, granulocyte macrophage colony stimulating factor (GM-CSF), insulin-like growth factor 1 (IGF-1), hepatocyte growth factor (HGF), and vascular endothelial growth factor (VEGF) was detected in patients with this disorder [57,58]. Some of them, e.g., GM-CSF, IL-1β, IL-6, IL-10, and TNF-α, are responsible for the expansion and activation of MDSCs [12,47]. There is evidence that oversynthesis of IL-6, TNF-α, metalloproteinases, and PGE promotes the adhesion of endometrial tissue to ectopic surfaces [54]. The factors mentioned above may be responsible for invasion and proliferation of endometrial implants to different locations, e.g., the sigmoid colon, rectum, ileum, appendix, bladder, ureter, and other, leading to extrapelvic endometriosis [59]. Extrapelvic EMS is an uncommon condition posing difficulties in early diagnosis and treatment. It may develop in the abdominal wall, pleura, diaphragm, and thorax. It has also been reported that implants can occur in the brain, conus medullaris, and lumbar vertebra [60]. However, the most uncommon and extremely rare case of extrapelvic implantation of endometriotic tissue is nasal endometriosis [61]. Although several theories of endometriosis development have been proposed, the pathogenesis of the condition is still unclear. It should be emphasized that none of the currently known theories can fully explain the different localizations of EMS. The most common theory, described by Sampson in 1927, highlights the role of retrograde menstruation, which leads to transplantation of endometrial cells and their implantation in the abdominopelvic cavity, including the uterosacral ligaments, bladder, pouch of Douglas, and retrocervix [62]. A result is the induction of local inflammatory response by implants formed of endometrial cells, which triggers a cascade of biological changes, including angiogenesis, anatomic distortion, fibrosis, adhesion scaring, and neuronal infiltration [60,61]. The Sampson theory explains the development of pelvic endometriosis. The Meyer theory seems to explain the distant sites of EMS better and it is based on spreading endometrial cells via hematogenous and/or lymphatic vessels, likewise cancer cells. Interestingly, Signorile and co-workers detected endometrial cells in lymph nodes [61,63].

Moreover, Nisolle et al. and Figueira et al. [64,65] have noted a predominant role of the base layer or bone marrow derivate stem cells in EMS pathogenesis, which are spread via fallopian tubes or vessels. The failure of the immune system may lead to implantation of endometrial tissue in different sites. Moreover, there is a hypothesis that endometriosis develops as a result of endometrial tissue dislocation from the uterine cavity during the process of organogenesis [61,63].

It is also possible to find implants in caesarean and other surgery scars, at the laparoscopic port site, umbilicus, thoracic cavity [66], rectus abdominis muscle, adductor compartment, or gluteal muscle [67].

It is known that women with EMS suffer from autoimmune inflammatory diseases. The development of extrapelvic endometriosis can be related to autoimmune etiology and factors promoting the growth of endometrial tissue at a distance from the uterus. Defects in the immunosurveillance in women suffering from endometriosis consistent with the autoimmune pathogenesis include increased polyclonal B-cell activity, abnormalities in the function and counts of T and B cells, presence of IgG, IgA, and IgM anti-endometrial autoantibodies in the serum, and reduced activity of NK cells [68]. Furthermore, immunosuppressive MDSCs are considered an important factor in the development of such autoimmune disorders as autoimmune hepatitis, diabetes type 1, IBD, MS, and RA. MDSCs are responsible for antigen-specific expansion of suppressive Tregs, which hamper T cell proliferation and non-specific T cell responses mediated by mitogen and may be involved in tissue inflammation, which appears during autoimmune diseases [69].

It should be emphasized that there are only a few published studies exploring MDSCs in patients suffering from endometriosis [7,8,9,48,70]. Using flow cytometry, Chen et al. (2017) and Zhang (2018) detected an increased percentage of MDSCs in the PB of patients vs. healthy women [7,9]. Gou et al. (2019) detected a higher number of M-MDSCs in the PB of patients with endometriosis compared to patients with dermoid cysts or uterine leiomyoma. The studies demonstrated that the high level of these cells was correlated with the presence of endometrial implants [70]. It has also been shown that MDSCs might inhibit proliferation of T cells and impair the function of CD4^+^/CD8^+^ effector T cells through synthesis of NO and, in this way, promote the progression of endometriosis [8,70]. Recently, a few reports have found that the population of monocytic MDSCs promotes progression of endometriosis in murine models [7,9]. These M-MDSCs produced high amounts of ROS and promoted differentiation and maturation of Tregs, which induce tolerance to endometrial implants. The authors also detected a higher level of ROS and NOX2 as well as p47phox mRNA responsible for ROS production by M-MDSCs in endometriosis patients. However, there was no correlation between the ROS level and the stage of the diseases [7]. Interestingly, higher numbers of both monocytic and polymorphonuclear MDSCs were detected in the peritoneal cavity of mice after transplantation of endometrial implants. This observation suggests that MDSCs may play an important role in the early stages of EMS development, and infiltration of MDSCs is critical for survival of endometrial tissue. Indeed, after depletion of MDSCs by the anti-Gr-1 monoclonal antibody, the endometriosis lesions were reduced in size and weight. Besides the higher percentage of PMN-MDSCs than M-MDSCs in the peritoneal cavity of mice, the M-MDSC population was more effective in inhibiting T cell proliferation [9].

Jiang and colleagues (2020) showed that mice with EMS had a higher level of ROS in both peritoneal cells and cells isolated from endometriotic lesions than mice without EMS. Their study showed that the blockade of the Notch pathway, which is a known regulator of angiogenesis, metastasis, and immunosuppressive TME in cancers, might lessen the percentage of MDSCs and the concentration of ROS. They speculate that the Notch signaling pathway may participate in ROS synthesis by MDSCs; however, further investigations are needed to confirm their hypothesis [48]. Other authors have demonstrated that the Notch pathway might promote the development of EMS through stimulation of angiogenesis of endometriotic lesions, strengthening the invasion of implants, and formation of fibrosis [48,71].

Recent reports have indicated accumulation of MDSCs in the peritoneal fluid of patients with endometriosis compared to PB or PF of patients with dermoid cysts or uterine leiomyoma [8,10,58]. Additionally, the levels of M-MDSCs in PF and PB were found to be significantly higher than those of PMN-MDSCs. An increased number of M-MDSCs were detected in more advanced stages of endometriosis, compared to the early stages of the disease [8]. It is well known that migration/accumulation of MDSCs can be induced by chronic inflammatory conditions, which involve such mediators as chemokines. It has been demonstrated that the CXCL1, CXCL2, and CXCL5 chemokines are involved in the migration of MDSCs to the peritoneal cavity [8,9]. It was shown in a murine model that intraperitoneal administration of CXCL1, CXCL2, and CXCL5 enhanced the migration of MDSCs. Elevated levels of these chemokines were observed in individuals with endometrial implants. It has been suggested that CXCL1 and CXCL2 released by macrophages may stimulate MDSC migration into the peritoneal cavity in early stages of endometriosis. Moreover, these chemokines can upregulate the immunosuppressive function of MDSCs [9]. Furthermore, elevated PF levels of chemokine CCL25 were found in patients with endometriosis and correlated with the stage of the disease and increased M-MDSC migration into the peritoneal cavity [10]. The CCL2 and its receptors (CCR2, CCR4, and CCR5) have been also reported to attract MDSCs in different disease models [70]. Other studies have shown that CCL5 and its receptor CCR5 are related to angiogenesis and migration of immune cells into the peritoneal cavity [70,72]. This phenomenon and the strong immunosuppressive activity of MDSCs against APCs, T cells, and NK cells suggest that this population may play an important role in the progression of endometriosis. Targeting CCR5^+^M-MDSCs can reduce the inflammatory condition and may constitute a new treatment option for EMS patients [70]. The use of CXCR2 inhibitors, such as SB265610 and SB225002, inhibited the migration of MDSCs both in vitro and in vivo in mice with endometrial implants. The use of anti-CCL25 or anti-CCR9 antibodies may also provide an effective therapeutic modality for patients with endometriosis [10]. Summarizing, the implantation and growth of endometrial tissue in the peritoneal cavity may be related to the local immunity in patients suffering from endometriosis, and MDSCs with CCR2/CCR5 expression may be a key driver in endometriosis progression [70].

## 5. Link between Endometriosis and Ovarian Cancer

Endometriosis is a chronic disease with a potential of malignant transformation. For the first time, the association between endometriosis and ovarian cancer was described by Sampson et al. in 1925 [73], but the questions of how ovarian endometriosis transforms into ovarian cancer and what subtypes of OC can develop in association with endometriosis have been considered for decades from biological, epidemiological, and clinical perspectives.

In 2016, Mogensen described a higher risk of ovarian cancer development in women with endometriosis versus the general population. Using histological type-specific analyses, they proved the increased risk of endometrioid and clear cell OC incidence in women suffering from EMS [74]. It should be stressed that studies of the transformation of endometriosis into OC have been carried out for years, but no explicit mechanisms have been fully described. However, it has been shown that endometrial implants share some similar features with ovarian cancer (Table 1).

Growing evidence supports the observation that multidimensional, molecular, genetic, and inflammatory mechanisms are involved in the transformation of endometriosis into EAOC. In both EMS and EAOC, early activation of the PI3K/AKT pathway has been described. It is also known that *PIK3CA* is involved in the early stages of endometriosis development. Additionally, *RUNX3* gene mutations have been described in women with ovarian endometriosis, and methylation of this gene has also been found in the early steps of EAOC development [80].

It is confirmed that the presence of endometriosis is associated with clear cell carcinoma, endometrioid ovarian cancer, and low grade serous carcinoma [81]. Especially atypical endometriosis that can be found in 1–3% of endometriotic cysts may be a precursor for clear cell carcinoma and endometrioid OC. Moreover, these two subtypes are diagnosed four times more often in women with endometriosis. The risk of transformation of endometriosis into OC was estimated at 0.7–5% [82].

Endometrioid ovarian cancer has solid and cystic patterns and is diagnosed at an earlier stage and younger age [81]. It constitutes approximately 10% of epithelial ovarian cancers. Mutations have been found in the following genes: *ARID1A, CTNNB1, *KMT2D**, *KMT2B, PIK3CA, PTEN,* and *PPR2R1A* [83]. The first of the genes mentioned above has a significant impact on tumor progression, since mutations in the *ARID1A* gene cause loss of functions in the tumor suppression mechanism. ARID1A, together with ARID1B, are Switch/Sucrose non-fermentable (SWI/SNF) protein family members. *ARID1A* encodes BRG-associated factor 250a (BAF250a or p270), which has an important role in cell proliferation and tumor suppression. The occurrence of the mutations in the *ARID1A* gene, the loss of function mutation, results in the loss of BAF250a protein expression. It has been shown that loss of BAF250a presumably occurs at an early stage in carcinogenesis, as observed in a subset of benign endometriotic cysts of the ovary and deep-infiltrating endometriosis [84].

Clear cell ovarian carcinoma (CCOC) can be described as glycogen-containing cells with clear cytoplasm and tubulocystic, solid, papillary, or mixed patterns [81]. It accounts for approximately 10% of the epithelial ovarian cancer group [85]. From the clinical point of view, CCOC has worse prognosis in all stages and ineffective therapeutic options when diagnosed at an advanced stage [85]. *ARID1A*, *PIK3CA*, *CTNNB1*, *CSMD3*, *LPHN3*, *LPR1B*, and *TP53* mutations are detected in CCOC [86]. The mutations in the listed genes are interesting from the potential therapeutic perspective. Boussios et al. describes the need of including sustained proliferative pathways, such as the PI3K/AKT/mTOR and the YES1/SRC tyrosine kinase pathways, or metabolic alterations, such as the glutathione biogenesis pathway, in *ARID1A*-deficient OCCC as the targets in future clinical trials. The synthetic agents targeting the *ARID1A* mutant setting are under investigation. Patients with *ARID1A* mutant ovarian clear cell carcinoma may benefit from inhibitors of the bromodomain and extraterminal domain (BET) family of proteins added to their treatment regimen. The mechanism of the enhanced sensitivity of *ARID1A* mutant cells to BET inhibitors may be explained by the reduced ARID1B expression, which is the effect of inhibition of the residual SWI/SNF complex [87].

Low-grade serous carcinoma (LGSC), which constitute less than 10% of ovarian serous carcinomas, have cells with small uniform nuclei and a micropapillary pattern with a low proliferative rate [81]. Even 70% of LGSOC are recognized at an advanced III–IV stage of the International Federation of Gynecology and Obstetrics (FIGO). Mutations have been found in two signaling pathways, i.e., PI3K/AKT/mTOR and KRAS/BRAF/MAPK. Clinically, LGSC can be considered as relatively chemoresistant [88].

The presence of atypical endometriosis can also be connected with the development of borderline ovarian tumors (BOTs) related to ovarian endometriosis, e.g., seromucinous, endometrioid, and clear cell borderline tumors [89].

Seromucinous borderline tumors (SMBTs) have papillary architecture with neutrophils-rimmed thick-walled cystic structures. They constitute ca. 4% of carcinomas and 7.6% of borderline tumors. SMBT are diagnosed in early stages and have similar prognosis to serous carcinomas [90]. Most of them have a good outcome. Endometriosis occurs in 30–70% of cases of SMBT, which frequently co-exists with endometriosis-associated cancers. It has been considered that SMBTs originate from atypical endometriotic cysts. From the genetic point of view, mutations in the *KRAS* (69%) gene and loss of *ARID1A* expression (33%) have been detected in the SMBT group [89].

Endometrioid borderline ovarian tumors (eBOTs) constitute approx. 0.2% of epithelial ovarian tumors and 2–10% of borderline tumors. They are diagnosed in the early stage and have rather good prognosis [91]. The presence of endometriosis was observed in 63% of patients. Furthermore, ca. 39% of patients had carcinoma or hyperplasia in the endometrium. In 50% of cases, benign endometrioid adenofibroma was detected. Mutation in the *KRAS* gene was detected in 29% of patients with associated endometriosis. Mutations in the *ARID1A* gene and anomalies in the PI3K/mTOR and Wnt/β-catenin pathways were found in the eBOT group [89].

Clear cell borderline tumors (CCBTs) are the most uncommon tumors in the borderline group with a prevalence rate constituting less than 1% of BOTs. They are also called atypical proliferative clear cell tumors. Their development is associated with atypical endometriosis [89]. Mesodermal (Müllerian) adenosarcomas, endometrioid stromal sarcomas, and carcinosarcomas are also recognized as neoplasms connected with ovarian endometriosis [92].

It has been well documented that chronic inflammation and an inflammatory milieu play a significant role in both endometriosis and ovarian cancer pathogenesis [93]. Furthermore, endometriosis and cancer-associated inflammation is connected with an elevated concentration of MDSCs [47], which may support the development and/or progression of endometriosis.

## 6. Conclusions

In conclusion, it is important to stress that most of the present knowledge of the activity of MDSCs has been provided by cancer studies. It has become clearly evident that one of the major consequences of MDSCs expansion is immunosuppression, angiogenesis, and release of cytokines or growth factors, which may stimulate the progression of endometriosis and cancers. Taking into account the immunosuppressive activity of MDSCs, e.g., the ability to suppress tumor-infiltrating lymphocyte activity, increase tumor-supportive M2-like macrophages, and regulate T cell responses [94], we hypothesize that they may also play an important role in the development of endometriosis-associated ovarian cancer. However, the aspect mentioned above requires further comprehensive research.

We hope that detailed elucidation of the mechanisms underlying the impaired immune responses by MDSCs in patients suffering from endometriosis may be a potential starting point for development of therapies to inhibit EMS progression and possibly prevent its transformation to EOC. However, further investigations are needed to shed some more light on this assumption.

## Figures and Tables

**Table 1 cells-10-00677-t001:** Similar features of endometriosis and ovarian cancer [2,5,75,76,77,78,79].

Characteristic	Endometriosis	Ovarian Cancer
Uncontrolled growth	✓	✓
Ability to invade other tissues	✓	✓
Proliferation via blood and lymphatic vessels	✓	✓
Neoangiogenesis	✓	✓
Lower ability to undergo apoptosis	✓	✓
Local inflammation	✓	✓
Increased volume of peritoneal fluid	✓	✓
Presence of LOH (loss of heterozygosity)	✓	✓
Mutation in genes *TP53*, *KRAS*, *PTEN*, *PIK3CA*, and *ARID1A*	✓	✓
High risk of return of the disease	✓	✓
Oral contraceptive pills—lower risk of development of the disease	✓	✓
Fallopian tube ligation—lower risk of development of the disease	✓	✓
Hysterectomy—lower risk of development of the disease	✓	✓
Multiple pregnancies—lower risk of development of the disease	✓	✓

## Data Availability

No new data were created or analyzed in this study. Data sharing is not applicable to this article.

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
