# Peer review of "The Role of Myeloid-Derived Suppressor Cells (MDSCs) in the Development and/or Progression of Endometriosis-State of the Art"

_cells, 2021, doi:10.3390/cells10030677_

Round 1

Reviewer 1 Report

thank you for giving me the opportunity to read this interesting article

the review is well conducted and describes exaustively state of arte of the role of MDSCs in the pathogenesis and progression of endometriosis

I would add the link between pelvic/extra-pelvic endometriosis and autoimmune disease and the possible mechanisms involved including suppressor cells (i.e.  10.1080/09513590.2019.1655727, https://doi.org/10.1016/j.jmig.2012.03.005, doi: 10.1093/humupd/dmz014)

Author Response

Reviewer 1

Dear Reviewer,

We would like to kindly thank you for your prompt and valuable review of our paper. We have addressed the comments raised by you and appropriate revision has been made in the manuscript. Point-by-point responses to the reviewer’s comments are listed below this letter.

Thank you again, we are truly hoping that now the paper will fulfill your requirements.

“Thank you for giving me the opportunity to read this interesting article. The review is well conducted and describes exhaustively state of arte of the role of MDSCs in the pathogenesis and progression of endometriosis. I would add the link between pelvic/extra-pelvic endometriosis and autoimmune disease and the possible mechanisms involved including suppressor cells.”

Response: Thank you very much for this important comment and suggestion. We have decided to add information concerning of the link between pelvic/extra-pelvic endometriosis and autoimmune disease and the possible mechanisms involved, including suppressor cells.

Page 7 lines 257-271

It has been shown that abnormally activated B lymphocytes may be involved in induction of autoimmune response in patients suffering from endometriosis [54]. In particular, antigen presenting cells (APC), which present endometrial self-antigens to autoreactive T and B cells, play an important role in this response. Interestingly, an increased level of anti‐endometrial antibodies has been detected in the serum of EMS patients [55,56]. Moreover, studies have shown that women with endometriosis have a greater risk of several autoimmune diseases than EMS-unaffected women [54,55]. Recent studies have demonstrated an association between endometriosis and increased risk of some autoimmune diseases, e.g. systemic lupus erythematosus (SLE), Sjogren's syndrome (SS), celiac disease (CLD), multiple sclerosis (MS), and inflammatory bowel disease (IBD). The presence of endometriosis may be also related to co-occurrence of  rheumatoid arthritis (RA), autoimmune thyroid diseases (ATD), and Addison’s disease [55]. However, it is unknown whether EMS is a risk factor or a consequence of these autoimmune disorders or whether they share the same mechanisms and biological pathways influencing their co-occurrence [55]. Therefore, further studies are recommended to explore this issue.

Page 8 lines 281-325

There is evidence that oversynthesis of IL-6, TNF-α, metalloproteinases, and PGE promotes the adhesion of endometrial tissue to ectopic surfaces [54]. The factors mentioned above may be responsible for invasion and proliferation of endometrial implants to different locations, e.g. the sigmoid colon, rectum, ileum, appendix, bladder, ureter, and other, leading to extrapelvic endometriosis [59]. Extrapelvic EMS is an uncommon condition posing difficulties in early diagnosis and treatment. It may develop in the abdominal wall, pleura, diaphragm, and thorax. It has also been reported that implants can occur in the brain, conus medullaris, and lumbar vertebra [60]. However, the most uncommon and extremely rare case of extrapelvic implantation of endometriotic tissue is nasal endometriosis [61]. Although several theories of endometriosis development have been proposed, the pathogenesis of the condition is still unclear. It should be emphasized that none of the currently known theories can fully explain the different localizations of EMS. The most common theory, described by Sampson in 1927, highlights the role of retrograde menstruation, which leads to transplantation of endometrial cells and their implantation in the abdominopelvic cavity, including the uterosacral ligaments, bladder, pouch of Douglas, and retrocervix [62]. A result is the induction of local inflammatory response by implantsformed of endometrial cells, which triggers a cascade of biological changes, including angiogenesis, anatomic distortion, fibrosis, adhesion scaring, and neuronal infiltration [60,61]. The Sampson theory explains the development of pelvic endometriosis. The Meyer theory seems to explain the distant sites of EMS better and it is based on spreading endometrial cells via hematogenous and/or lymphatic vessels, likewise cancer cells. Interestingly, Signorile and co-workers detected endometrial cells in lymph nodes [61,63].

Moreover, Nisolle et al. and Figueira et al. [64,65] have noted a pre-dominant role of the base layer or bone marrow derivate stem cells in EMS pathogenesis, which are spread via fallopian tubes or vessels. The failure of the immune system may lead to implantation of endometrial tissue in different sites. Moreover, there is a hypothesis that endometriosis develops as a result of endometrial tissue dislocation from the uterus during the process of organogenesis [61,63].

It is also possible to find implants in caesarean and other surgery scars, at the laparoscopic port site, umbilicus, thoracic cavity [66], rectus abdominis muscle, adductor compartment, or gluteal muscle [67].

It is known that women with EMS suffer from autoimmune inflammatory diseases. The development of extrapelvic endometriosis can be related to autoimmune aetiology and factors promoting the growth of endometrial tissue at a distance from the uterus. Defects in the immunosurveillance in women suffering from endometriosis consistent with the autoimmune pathogenesis include increased polyclonal B-cell activity, abnormalities in the function and counts of T and B cells, presence of IgG, IgA, and IgM anti-endometrial autoantibodies in the serum, and reduced activity of NK cells [68]. Furthermore, immunosuppressive MDSCs are considered an important factor in the development of such autoimmune disorders as autoimmune hepatitis, diabetes type 1, IBD, MS, and RA. MDSCs are responsible for antigen-specific expansion of suppressive Tregs, which hamper T cell proliferation and non-specific T cell responses mediated by mitogen and may be involved in tissue inflammation, which appears during autoimmune diseases [69].

Dear Reviewer,

We have made relevant changes and revisions to the original manuscript to address the recommendations of the reviewers. We believe that the revisions that we were able to conduct address all major comments and have significantly improved the quality of our manuscript.

Please address these concerns.

Best regards,

Dorota Suszczyk and co-authors

Reviewer 2 Report

In this article, Suszczyk D et al review the impact of MDSCs on different populations of immune cells, focusing on their immunosuppressive role in the immune system, which may be related with the pathogenesis and/or progression of endometriosis and its transformation into ovarian cancer. The manuscript is straightforward, well written, and concise, and has clear results, within the scope of a review article. Definitely deserves to be published and is a valuable contribution to the “cellsjournal. Some minor flaws need to be addressed before publication.

Minor points:

[1] “2. Immunophenotype of myeloid-derived suppressor cells (MDSCs)”, Page 2/11:

The presence of MDSCs has an impact on the efficacy of immunotherapies and patient outcomes.”.

Moreover, among other mechanisms of angiogenesis-directed immune tolerance is included the accumulation of immunoregulatory cells (Treg cells, myeloid-derived suppressor cells). Based on that, combination of immunotherapy with VEGF blockade is a therapeutic approach that potentially increase the anti-tumor activity of immunotherapy.

Recommended reference: Demircan NC, et al. Current and future immunotherapy approaches in ovarian cancer. Ann Transl Med. 2020;8(24):1714.

[2] “5. Link between endometriosis and ovarian cancer”, Pages 7/11 and 8/11:

Endometrioid ovarian cancer has solid and cystic patterns and is diagnosed at an earlier stage and younger age [66]. It constitutes approximately 10% of epithelial ovarian cancers. Mutations have been found in the following genes: ARID1A, CTNNB1, KMT2D, KMT2B, PIK3CA, PTEN, and PPR2R1A [68].”.

At that point, please do report that mutations in ARID1A result in the loss of BRG-associated factor 250a (BAF250a), a protein with an important role in cell proliferation and tumor suppression. It has been shown that loss of BAF250a presumably occurs at an early stage in carcinogenesis, as has been observed in a subset of benign endometriotic cysts of the ovary and deep-infiltrating endometriosis.

Recommended reference: Samartzis EP, et al. Endometriosis-associated ovarian carcinomas: insights into pathogenesis, diagnostics, and therapeutic targets-a narrative review. Ann Transl Med. 2020;8(24):1712.

[3] 5. Link between endometriosis and ovarian cancer”, Page 8/11:

Clear cell ovarian carcinoma (CCOC) can be described as glycogen-containing cells with clear cytoplasm and tubulocystic, solid, papillary, or mixed patterns [66]. It accounts for approximately 10% of the epithelial ovarian cancer group [69]. From the clinical point of view, CCOC has worse prognosis in all stages and ineffective therapeutic options when diagnosed at an advanced stage [69]. ARID1A, PIK3CA, CTNNB1, CSMD3, LPHN3, LPR1B, and TP53 mutations are detected in CCOC [70].”.

From the therapeutic perspective, targeting of sustained proliferative pathways, such as the PI3K/AKT/mTOR and the YES1/SRC tyrosine kinase pathways, or metabolic alterations, such as the glutathione biogenesis pathway, in ARID1A-deficient ovarian clear cell carcinoma should be considered in future clinical trials. Such synthetic lethal agents in the ARID1A mutant setting are currently in clinical development. The inhibitory effects on residual SWI/SNF function, specifically via reduced ARID1B expression, may explain the enhanced sensitivity of ARID1A mutant cells to bromodomain and extraterminal domain (BET) inhibitors. As such, patients with ARID1A mutant ovarian clear cell carcinoma may benefit from inhibitors of the BET family of proteins added to their treatment regimen. Please, do incorporate this information.

Recommended reference: Boussios S, et al. Wise Management of Ovarian Cancer: On the Cutting Edge. J Pers Med. 2020;10(2):41.

[4] A workflow diagram would be of benefit for the readers.

Author Response

Reviewer 2

Dear Reviewer,

We would like to kindly thank you for your prompt and valuable review of our paper. We have addressed the comments raised by you and appropriate revision has been made in the manuscript. Point-by-point responses to the reviewer’s comments are listed below this letter.

Thank you again, we are truly hoping that now the paper will fulfill your requirements.

In this article, Suszczyk D et al review the impact of MDSCs on different populations of immune cells, focusing on their immunosuppressive role in the immune system, which may be related with the pathogenesis and/or progression of endometriosis and its transformation into ovarian cancer. The manuscript is straightforward, well written, and concise, and has clear results, within the scope of a review article. Definitely deserves to be published and is a valuable contribution to the “Cells” journal. Some minor flaws need to be addressed before publication. Minor points:

[1] “2. Immunophenotype of myeloid-derived suppressor cells (MDSCs)”, Page 2/11:

“The presence of MDSCs has an impact on the efficacy of immunotherapies and patient outcomes.”.

Moreover, among other mechanisms of angiogenesis-directed immune tolerance is included the accumulation of immunoregulatory cells (Treg cells, myeloid-derived suppressor cells). Based on that, combination of immunotherapy with VEGF blockade is a therapeutic approach that potentially increase the anti-tumor activity of immunotherapy.

Recommended reference: Demircan NC, et al. Current and future immunotherapy approaches in ovarian cancer. Ann Transl Med. 2020;8(24):1714.

Response: Thank you very much for this important comment and suggestion. We have included this information and the recommended reference in the revised manuscript. We agree that it will help to ensure a higher quality of our manuscript and make it more interesting.

Page 2 lines: 78-86

The latest literature emphasizes the role of vascular endothelial growth factor A (VEGF-A). Vascular endothelial growth factor A (VEGF-A) secreted by cancer cells not only induces angiogenesis but also causes immunosuppression. VEGF blocks DC maturation causing decreased antigen presentation to T cells. This mechanism of angiogenesis-directed immune tolerance includes accumulation of immunoregulatory cells, e.g. Treg cells and MDSCs, and inhibition of T cell differentiation, proliferation, and functions. Since VEGF has such a great impact on angiogenesis and immunosuppression, therapy with VEGF blockade is another potential method to increase the anti-tumor activity of immunotherapy[19].

[2] “5. Link between endometriosis and ovarian cancer”, Pages 7/11 and 8/11:

“Endometrioid ovarian cancer has solid and cystic patterns and is diagnosed at an earlier stage and younger age [66]. It constitutes approximately 10% of epithelial ovarian cancers. Mutations have been found in the following genes: ARID1A, CTNNB1, KMT2D, KMT2B, PIK3CA, PTEN, and PPR2R1A [68].”.

At that point, please do report that mutations in ARID1A result in the loss of BRG-associated factor 250a (BAF250a), a protein with an important role in cell proliferation and tumor suppression. It has been shown that loss of BAF250a presumably occurs at an early stage in carcinogenesis, as has been observed in a subset of benign endometriotic cysts of the ovary and deep-infiltrating endometriosis.

Recommended reference: Samartzis EP, et al. Endometriosis-associated ovarian carcinomas: insights into pathogenesis, diagnostics, and therapeutic targets-a narrative review. Ann Transl Med. 2020;8(24):1712.

Response: We are grateful to the Reviewer for raising this important issue. In the revised version of the manuscript, we have included information about the role of the loss of BRG-associated factor 250a (BAF250a), a protein with an important role in cell proliferation and tumor suppression.

Page 12 lines 436-445

The first of the genes mentioned above has a significant impact on tumor progression, since mutations in the ARID1A gene cause loss of functions in the tumor suppression mechanism. ARID1A, together with ARID1B, are Switch/Sucrose non-fermentable (SWI/SNF) protein family members. ARID1A encodes BRG-associated factor 250a (BAF250a or p270), which has an important role in cell proliferation and tumor suppression. The occurrence of the mutations in the ARID1A gene, the loss of function mutation, results in the loss of BAF250a protein expression. It has been shown that loss of BAF250a presumably occurs at an early stage in carcinogenesis, as observed in a subset of benign endometriotic cysts of the ovary and deep-infiltrating endometriosis [84].

[3] “5. Link between endometriosis and ovarian cancer”, Page 8/11:

“Clear cell ovarian carcinoma (CCOC) can be described as glycogen-containing cells with clear cytoplasm and tubulocystic, solid, papillary, or mixed patterns [66]. It accounts for approximately 10% of the epithelial ovarian cancer group [69]. From the clinical point of view, CCOC has worse prognosis in all stages and ineffective therapeutic options when diagnosed at an advanced stage [69]. ARID1A, PIK3CA, CTNNB1, CSMD3, LPHN3, LPR1B, and TP53 mutations are detected in CCOC [70]”.

From the therapeutic perspective, targeting of sustained proliferative pathways, such as the PI3K/AKT/mTOR and the YES1/SRC tyrosine kinase pathways, or metabolic alterations, such as the glutathione biogenesis pathway, in ARID1A-deficient ovarian clear cell carcinoma should be considered in future clinical trials. Such synthetic lethal agents in the ARID1A mutant setting are currently in clinical development. The inhibitory effects on residual SWI/SNF function, specifically via reduced ARID1B expression, may explain the enhanced sensitivity of ARID1A mutant cells to bromodomain and extraterminal domain (BET) inhibitors. As such, patients with ARID1A mutant ovarian clear cell carcinoma may benefit from inhibitors of the BET family of proteins added to their treatment regimen. Please, do incorporate this information.

Recommended reference: Boussios S, et al. Wise Management of Ovarian Cancer: On the Cutting Edge. J Pers Med. 2020;10(2):41.

Response: Thank you for this comment. We have provided this information in the revised manuscript, which in our opinion will increase its quality.

Page 13 lines 451-461

The mutations in the listed genes are interesting from the potential therapeutic perspective. Boussios et al. describes the need of including sustained proliferative pathways, such as the PI3K/AKT/mTOR and the YES1/SRC tyrosine kinase pathways, or metabolic alterations, such as the glutathione biogenesis pathway, in ARID1A-deficient OCCC as the targets in future clinical trials. The synthetic agents targeting the ARID1A mutant setting are under investigation. Patients with ARID1A mutant ovarian clear cell carcinoma may benefit from inhibitors of the bromodomain and extraterminal domain (BET) family of proteins added to their treatment regimen. The mechanism of the enhanced sensitivity of ARID1A mutant cells to BET inhibitors may be explained by the reduced ARID1B expression, which is the effect of inhibition of the residual SWI/SNF complex [87].

 [4] A workflow diagram would be of benefit for the readers.

Response: Thank you for this valuable remark. We have decided to add a workflow diagram to the manuscript to make it more readable for the readers and increase its overall comprehensibility.

Dear Reviewer,

We have made relevant changes and revisions to the original manuscript to address the recommendations of the reviewers. We believe that the revisions that we were able to conduct address all major comments and have significantly improved the quality of our manuscript.

Please address these concerns.

Best regards,

Dorota Suszczyk and co-authors
